# Altered Gut Microbiota Composition Is Associated with Difficulty in Explicit Emotion Regulation in Young Children

**DOI:** 10.3390/microorganisms11092245

**Published:** 2023-09-06

**Authors:** Hideaki Fujihara, Michiko Matsunaga, Eriko Ueda, Takamasa Kajiwara, Aya K. Takeda, Satoshi Watanabe, Kairi Baba, Keisuke Hagihara, Masako Myowa

**Affiliations:** 1Graduate School of Education, Kyoto University, Yoshida-honmachi, Sakyo-ku, Kyoto 606-8501, Japan; hidemontanabozeman@gmail.com (H.F.);; 2Japan Society for the Promotion of Science, Chiyoda-ku, Tokyo 102-0083, Japan; 3Department of Advanced Hybrid Medicine, Graduate School of Medicine, Osaka University, 2 Chome-2 Yamadaoka, Suita 565-0871, Japan; 4Cykinso, Inc., 1-36-1 Yoyogi, Shibuya-ku, Tokyo 151-0053, Japan

**Keywords:** gut microbiome, executive function, explicit emotion regulation, dietary habits, preschoolers

## Abstract

Executive function (EF) consists of explicit emotion regulation (EER) and cognitive control (CC). Childhood EER in particular predicts mental and physical health in adulthood. Identifying factors affecting EER development has implications for lifelong physical and mental health. Gut microbiota (GM) has attracted attention as a potential biomarker for risk of physical and mental problems in adulthood. Furthermore, GM is related to brain function/structure, which plays a crucial role in emotional processing. However, little is known about how GM compositions are associated with the development of emotion regulation in early childhood. Therefore, in this study, we examined 257 children aged 3–4 to investigate links between GM and risk to EF. EF was measured using the Mother-Reported Behavior Rating Inventory of Executive Function–Preschool version. GM composition (alpha/beta diversity and genus abundance) was evaluated using 16S rRNA gene sequencing and compared between EF-risk and non-risk groups. Our results show that children with EER-risk (an index of inhibitory self-control) had a higher abundance of the genera *Actinomyces* and *Sutterella*. Although we have not established a direct link between GM and CC risk, our findings indicate that GM of preschoolers is closely associated with emotional processing and that EERrisk children have more inflammation-related bacteria.

## 1. Introduction

Executive function (EF) is a cognitive function defined as the ability to control and coordinate behavior and thoughts toward one’s own goals [1]. It undergoes remarkable development between 3 and 4 years of age [2]. Childhood EF has commanded much attention since it was identified as a predictive factor of wealth and physical health in adulthood [3].

EF consists of two distinct domains: “cognitive control” (CC) and “explicit emotion regulation” (EER). CC is the ability to solve abstract problems and is oriented toward the achievement of an objective without emotional context (i.e., *Cool* EF [4]), while EER reflects the ability to exert control over one’s emotional state (i.e., *Hot* EF [5,6]). Several large, prospective cohort studies suggest that EER is closely related to later-life outcomes; EER in early childhood predicts problem behaviors (e.g., anxiety tendencies and aggression) during school years [7]. Moreover, childhood EER predicts economic, mental, and physical health, and brain aging (i.e., the difference between participants’ predicted age from MRI data and their exact chronological age) in middle age [8].

Although adequate support for the development of EER from early childhood is therefore crucial, previous support and intervention methods have not always proved effective for all children, reflecting large individual differences [9,10,11]. To overcome such limitations, we need to identify biological factors underlying individual differences in the development of EER in early childhood.

This study focuses on the childhood gut microbiome as a potential biomarker relevant to individual differences in EER development. Intestinal microbiome composition is known to be associated not only with physical diseases (e.g., diabetes [12]) but also with psychological disorders (e.g., depression and anxiety [13]). The gut microbiome affects brain functions through autonomic nerves, neuropeptides, hormones, and the immune system [14]. Animal studies have shown that disruption of the microbiota by antibiotic treatment modulates neural activity in limbic regions (hypothalamus and hippocampus) and causes social stress (e.g., anxiety [15]).

Importantly, the microbiome changes rapidly during infancy and reaches an adult-like configuration at around 3–5 years after birth [16,17]. The first 5 years of life are a “sensitive period” that largely determines the lifetime gut microbiome composition of each individual. It seems possible that individual differences in childhood EER might be associated with stabilization of the gut microbiome composition during this developmental period.

In early childhood, the prefrontal cortex is still in its maturation stage [18,19], making it challenging to exert top-down control over the activation of the limbic system in response to emotional stimuli [20]. Recent studies have elucidated the relationship between the gut microbiome and brain structure/function related to such emotion regulation. For example, Gao et al. [21] reported that microbiome diversity during the first year of life is associated with the activation of functional brain networks of the amygdala-thalamus and insula-anterior cingulate cortex (ACC), which play a critical role in emotional processing. Carlson et al. [22] found that relative abundance of the *Streptococcus* genus in the first month of life is associated with the volume of the amygdala. A study in adults also showed that probiotic interventions reduced insular and somatosensory cortical activities in stressful situations [23], implying a close association between gut microbiota and limbic system activities involved in emotion processing.

It remains unknown whether and how microbiome composition in the early stages of life relates to psychological and behavioral phenotypes. Loughman et al. [24] found that decreased *Prevotella* abundance at 12 months of age is associated with emotional problems at 2 years of age (e.g., anxiety and depression). Moreover, infant gut microbiome is associated with temperament at 6 months of age [25] and language development at 2 years of age [26]. 

These studies, which focused on the period before the gut microbiome of an individual stabilizes, suggest links between microbiota and psychological functioning. However, as mentioned, the intestinal microbiome does not stabilize until the age of 3–5 years [16]. It is important to bear in mind that brain and psychological functioning shows remarkable flexibility and undergoes dramatic changes during the sensitive period when the gut microbiome composition is taking shape. EER develops between the ages of 3–4 years, which coincides with the time when an individual’s gut microbiome reaches adult levels and stabilizes. Gut microbiome composition during this period might therefore be associated with the individual variability in EER development.

This study investigated the relationship between gut microbiome and the development of EF in 257 children aged 3–4 years. We hypothesized that links with the gut microbiota would be found especially in the EER rather than the CC. There were two reasons for this hypothesis. Prior studies have indicated that gut microbiota is associated with brain function and structures involved in emotional processing, particularly the limbic system [21,22,23]. Second, early childhood is a developmental stage during which the prefrontal cortex is still immature, making it difficult to regulate the activation of the limbic system through top-down prefrontal control [18,19,20]. When the relationship between EF risk and gut microbiota was observed, we conducted a post hoc analysis to exploratively examine other potential factors that children with EF risk might have. In this study, particularly considering the association with gut microbiota, we focused on dietary habits [27,28], inflammatory diseases, and physical symptoms [29,30], conducting comparisons between the EF-risk and non-risk groups. 

## 2. Materials and Methods

### 2.1. Participants and Procedure

We collected data on 539 Japanese children aged 0–4 years between December 2020 and March 2021 as part of a research project titled A large-scale study of intestinal bacterial flora, psychophysiological, and cognitive and behavioral characteristics in infants and their mothers. Participants were recruited through nursery schools throughout Japan who agreed to cooperate in the study. We obtained written informed consent from the children’s mothers. Of the original 539 participants, in this study, we analyzed data from those who met the following criteria: (1) were between 3 and 4 years old (and who completed the executive function questionnaire BRIEF-P, see Section 2.2.1), (2) who completed both stool collection and all questionnaires at home, and (3) the mother was not suffering from depression (assessed by the Beck Depression Inventory-II (BDI-II)) (Figure 1). Finally, we analyzed 257 children’s data (mean age = 46.5 months, SD = 6.3 months, range 36–57 months). The study was approved by the Medical Ethics Committee of Kyoto University (no. R2624) and registered in the UMIN system (UMIN000043945).

### 2.2. Questionnaires

#### 2.2.1. BRIEF-P

We assessed the children’s EF using the Japanese version of the Behavior Rating Inventory of Executive Function–Preschool version (BRIEF-P). This is a questionnaire in which parents report their preschool children’s daily EF behavior [31,32]. It consists of 63 items concerning problematic behaviors and uses a 3-point rating scale (1 = never, 2 = sometimes, 3 = often). This questionnaire yields a score for each of five subscales: *inhibition*, *emotion control*, *working memory*, *shifting*, and *planning* (Appendix A). Following Gioia et al. [31] and Sherman and Brooks [33], we collapsed the subscales into the following three broad indices: *inhibitory self-control* (ISC), *flexibility* (FL), and *emergent metacognition* (EM). 

EER, the ability to control one’s emotional state, corresponds to the broad indices ISC and FL of BRIEF-P [34,35]. ISC indicates the child’s ability to regulate emotions and behaviors according to environmental demand and is based on the subscale scores of *inhibition* and *emotion control* (e.g., [31]). FL shows the ability to switch emotions and behaviors to appropriate situations and is based on the subscale scores of *emotion control* and *shifting* (e.g., [31,33]). Both ISC and FL include *emotion control*, which specifically assesses the child’s ability to regulate emotional responses [31] and is a core component of EER. The broad index EM corresponds to CC in reflecting the capacity for cognitive control [34,35]. It concerns the child’s ability to solve problems through planning while holding ideas and motivation to complete the task in *working memory* (Figure 1). 

As a SD of 1.5 or higher than the mean score is recommended as the cutoff value for abnormality (risk) in the original questionnaires [33,36], we used cutoff values of 1.791, 1.731, and 1.654 points, for ISC, EM, and FL, respectively (see Appendix A for more information about the BRIEF-P mean scores for all participants, the risk group, and the non-risk group, respectively). Children who exceeded the cutoff value in the target index were assigned to the corresponding risk group. Then, as some children were at risk for e.g., both ISC and EM, or for all three domains, we included all but the target at-risk children in the corresponding non-risk group. For example, the ISC non-risk group included children at risk for FL (N = 4), EM (N = 8), or both FL and EM (N = 3) (Appendix A). The detailed breakdown of the type of risk and the number of children in each ISC, FL, and EM risk/non-risk group is shown in Appendix A. In the main analysis of this study, we compared gut microbiota composition between the EF and non-risk groups. Furthermore, we established a control group (N = 216) comprising only individuals who did not correspond to any risks (Appendix A). By comparing gut microbiota composition between the EF risk and control groups, we confirmed that there were no discrepancies in the results of the main analysis. Any missing values were replaced with 1, and three participants with more than 12 missing values among the 63 items were excluded from the study (following [31,37]).

#### 2.2.2. Dietary Habits (Food Intake Frequency, JDI Score, Picky Eating)

We asked the child’s mother how often her child consumed any of the 24 food items in the past week on a 5-point scale (never eaten = 1; 1–3 times = 2; 4–6 times = 3; 1 time every day = 4; more than two times every day = 5). We also asked whether the child had picky eating behavior, using a yes/no alternative response. Picky eaters are defined as children who consume an inadequate variety of foods by rejecting foods that are familiar or unfamiliar to them [38]. We also calculated the Japanese Dietary Index (JDI) score to investigate the relationship between EF risk and gut microbiome characteristics. Previous research found that the Japanese diet was associated with a reduced risk of dementia [39]. Intervention studies for adults also showed that consuming a Japanese diet for one month could alter the gut microbiota and improve physical health (e.g., weight loss) [40]. 

The formula for calculating the JDI score was as follows [39]: JDI score = unrefined grains + green and yellow vegetables + fruits + soy products + natto + beans + mushrooms + pickled vegetables + seaweed + seafood − meat. Usually, the JDI score includes coffee intake as a beneficial dietary item, but it was not included in this study because the frequency of coffee consumption was not known. For each food item, a score of 1 was assigned for above-median frequency of intake and 0 for below-median frequency. For any missing values, we applied predictive mean matching [41,42]. The JDI scores in this study ranged from 0 to 11; higher scores indicate a healthier Japanese diet. 

#### 2.2.3. Physical Symptoms

We collected data on the frequency of diarrhea, the prevalence of allergic symptoms, and feces state. Diarrhea frequency was measured on a 5-point scale (none = 1; once a week = 2; once every 4–5 days = 3; once every 2–3 days = 4; every day = 5). Regarding allergy symptoms, mothers answered Yes or No according to whether the child had any of the following symptoms: urticaria/atopy, asthma, allergic rhinitis, hay fever, or food allergy. For assessment of the feces state, we used the Bristol Stool Scale (see Appendix A) [43], in which scores 1 and 2 indicate constipation, scores 3–5 may be considered as ‘within the range of normal defecation’, and scores 6 and 7 indicate diarrhea [43]. Following previous studies [43,44], we considered scores 1, 2, 6, and 7 as abnormal.

#### 2.2.4. BDI-II

We assessed maternal depression using the Japanese version of the BDI-II [45,46,47] Several studies have indicated that mothers with depression tend to overestimate their children’s problem behaviors [48,49]. Therefore, to exclude potential bias, we excluded data provided by mothers with moderate or severe depression (i.e., cutoff score 20 or higher) from the analysis (N = 40). 

### 2.3. Demographic Data

We collected demographic information about each child’s birth date, sex, antibiotic treatment within the past 3 months, weight (kg), height (cm), family income, mother’s years of schooling, number of siblings, and delivery mode (vaginal delivery and cesarean section). BMI was calculated using the formula: weight/height^2^ (kg/m^2^).

### 2.4. Fecal Sampling, DNA Extraction, and Sequencing

Fecal samples were collected at home using Mykinso fecal collection kits containing guanidine thiocyanate solution (Cykinso, Tokyo, Japan), transported at ambient temperature, and stored at 4 °C. DNA extraction from fecal samples was performed using an automated DNA extraction machine (GENE PREP STAR PI-480, Kurabo Industries Ltd., Osaka, Japan) following the manufacturer’s protocol. Children who failed to provide stool samples were excluded from the analysis (N = 9). Detailed sequencing methods are described in Watanabe et al. [50]. Briefly, amplicons of the V1V2 region were prepared using the forward primer (16S_27Fmod: TCG TCG GCA GCG TCA GAT GTG TAT AAG AGA CAG AGR GTT TGA TYM TGG CTC AG) and the reverse primer (16S_338R: GTC TCG TGG GCT CGG AGA TGT GTA TAA GAG ACA GTG CTG CCT CCC GTA GGA GT). The libraries were sequenced in a 250-bp paired-end run using a MiSeq Reagent Kit v2 (Illumina; 500 cycles).

### 2.5. Taxonomy Assignment Based on 16S rRNA Gene Sequencing

Data processing and assignment using the QIIME2 pipeline (version 2020.8) [51] were performed as follows: (1) joining paired-end reads, filtering, and denoising with a divisive amplicon denoising algorithm (DADA2) and (2) assigning taxonomic information to each amplicon sequence variant (ASV) using a naive Bayes classifier in the QIIME2 classifier. The classifier was trained using a robust taxonomy simplifier for SILVA (arts-SILVA), which was originally developed from the 16S rRNA taxonomy dataset based on SILVA 138 [52]. arts-SILVA was developed for the purpose of making Mykinso testing reports easier to understand for those who are unfamiliar with the complex rules of taxonomic nomenclature [53]. arts-SILVA simplifies the resulting taxonomic assignments by removing study-related labels, curating notable misentries, and generalizing uncommon names in the SILVA database. To obtain arts-SILVA, the V1V2 regions of the SILVA reference sequences were extracted and clustered according to the original manuscript for QIIME2 preparation in SILVA. Subsequently, some unnecessary/seemingly miss-labeled entries were deleted (i.e., we removed labels with little information such as “D_6__unclutured bacteria”) and duplicate entries such as “D_0__Bacteria;D_1__Bacteria Firmicutes” were corrected to “D_0__Bacteria;D_1__Firmicutes”). Subsequently, unnecessary taxa such as “D_6__human metagenome” were removed by manual inspection, and a consensus taxonomy was assigned to each cluster for which 100% of the assigned taxa were in 100% agreement. Finally, the label “Ambiguous taxa” was removed. 

### 2.6. Diversity Analysis

We measured alpha diversity at the ASV level using the Shannon, Chao1, Faith’s phylogenetic diversity (PD), and observed species indexes. The Shannon index is a measure of richness and evenness. Chao1 estimates the total number of ASVs that would be observed with infinite sampling. Faith’s PD is a phylogenetic measure of taxon richness and is expressed as the number of tree units observed in the sample. Finally, the observed species was simply the observed number of ASVs per sample. Alpha diversity metrics were calculated with QIIME 2′s “diversity” plugin. We also used beta diversity to evaluate differences in community composition between samples using the Bray–Curtis distance method.

### 2.7. Statistical Analysis

We performed all data manipulation, analyses, and graphics using RStudio (version 4.1.0) [54] and used microbiome R for all analyses. We used the R packages ANCOM-BC (v 1.4.0) [55], patchwork (v1.1.1), ggpubr (v0.4.0), and gglplot2 (v3.3.6) for visualization. The R package phyloseq (v1.38.0) [56] and tidyverse (v1.3.2) were used for data handling. Permutation-based multivariate analysis of variation (PERMANOVA) was performed using the ‘vegan’ package (Adonis, v 2.6.2). To correct for multiple hypothesis testing, *p* values were adjusted based on the false discovery rate (FDR) using the q-values method. The q-values were obtained using the qvalue function (v 2.26.0) of the qvalue Bioconductor R package (http://bioconductor.org/packages/release/bioc/html/qvalue.html, accessed on 25 August 2022). We set the significance level at *p* < 0.05 and *q* < 0.10 [57].

#### 2.7.1. Gut Microbiota Diversity

To compare alpha diversity metrics (the Shannon, Chao1, Faith’s PD, and observed species indexes) between the EF-risk and non-risk groups, we conducted an ANCOVA (package rstatix in R v 0.7.0). The distance between samples was measured using the Bray–Curtis distance matrix, and principal coordinate analysis (PCoA) was used to visualize the matrix in a 2D plot, where each dot represents the entire microbiome of a single sample. Differences in Bray–Curtis distance between groups were assessed using a nonparametric PERMANOVA test using the ‘vegan’ package, with 999 permutations. Here, we included antibiotic treatment within the past 3 months (i.e., before stool sampling) and sex as covariates, given their influence on gut microbiome composition [25,58,59].

#### 2.7.2. Gut Microbiota Composition

Differential abundance at the genus level was analyzed using the R package ANCOM-BC [55]. The challenge in differential abundance analysis is the compositional nature of microbiome data depending on sampling and sequencing depth (the number of reads assigned to an ASV must be interpreted in relation to the total number of reads for that sample). Analysis of microbiome compositions with bias correction (ANCOM-BC) addresses the compositional nature of the data by estimating and then eliminating the bias introduced by differences in sampling fractions in the observed counts. This methodology uses relative abundance to infer absolute abundance while controlling the FDR and deals with excess zeros by incorporating the ANCOM-II procedure [60]. We conducted prevalence filtering to obtain more robust results in ANCOM-II [61]; taxa had to appear in at least 25% of the samples with a relative abundance of at least 0.001% to be included in ANCOM-BC [62]. Fifty-nine genera were included in the comparison between the EF-risk and non-risk groups (Appendix A). In this analysis, we also included antibiotic treatment within the past 3 months and sex as covariates.

#### 2.7.3. Dietary Habits

For analysis of food consumption frequencies, we ran the Brunner–Munzel test [63] to compare the risk group with the non-risk group, as normality and homoscedasticity were not met. To compare JDI scores, we used the Mann–Whitney *U* test as normality was not met. Finally, for comparisons of the percentage of picky eaters (i.e., categorical variable), we performed a Fisher’s exact test.

#### 2.7.4. Physical Symptoms

First, we conducted the Mann–Whitney *U* test to compare diarrhea frequencies between the EF-risk and EF non-risk groups, as normality was not met. Second, Fisher’s exact test was performed to compare the percentage of abnormal stool forms and allergic symptoms (i.e., categorical variable).

#### 2.7.5. Demographic Data

See Appendix A for details on the statistical methods used for demographic data.

## 3. Results

### 3.1. Microbiome Components in Inhibitory Self-Control (ISC)-Risk and Non-Risk Groups

According to the cutoff value of the ISC index, 26 and 231 children were respectively allocated to the ISC-risk group (mean age = 46.04 ± 6.32 months, 19 boys) and ISC non-risk group (mean age = 46.59 ± 6.34 months, 133 boys). Regarding the demographic characteristics age, sex, BMI, family income, mother’s years of schooling, siblings, and birth mode, there were no group differences (Appendix A). First, we compared the alpha diversity metrics (Shannon, Chao1, Faith’s PD, and observed species index) between the ISC-risk and the ISC non-risk groups using analysis of covariance (ANCOVA). No significant differences were found in alpha diversity metrics (Figure 2a–d, Appendix A for details). Next, we performed PERMANOVA to uncover differences in the structure of gut microbiota between two groups and found no group difference in beta diversity (Figure 2e, Appendix A). Furthermore, we conducted ANCOM-BC to compare bacterial genera abundance between the two groups. The ISC-risks had a significantly higher relative abundance of *Actinomyces* (*p* < 0.001, *q* = 0.006) and *Sutterella* (*p* = 0.003, *q* = 0.084, Figure 2f,g, and see Appendix A for details).

### 3.2. Microbiome Components in the Flexibility (FL)-Risk and Non-Risk Groups

According to the cutoff value of the FL index, 18 and 239 children were allocated to the FL-risks (mean age = 44.56 ± 6.46 months, 15 boys) and the FL non-risk (mean age = 46.68 ± 6.31 months, 137 boys), respectively. Concerning demographic characteristics, FL non-risk group children were significantly more likely to have siblings (Appendix A). Regarding the microbiota diversity metrics, there were no differences between the FL-risk and the FL non-risk groups (Figure 3a–e, Appendix A). ANCOM-BC revealed no significant differences between the two groups (Appendix A).

### 3.3. Microbiome Components in the Emergent Metacognition (EM)-Risk and Non-Risk Groups

According to the cutoff value of the EM index, 20 and 237 children were respectively allocated to the EM-risk group (mean age = 45.55 ± 6.40 months, 14 boys) and the EM non-risk group (mean age = 46.62 ± 6.33 months, 138 boys). There were no differences in demographic characteristics between the groups (Appendix A). Furthermore, the two groups did not differ in terms of the microbiota diversity metrics (Figure 4a–e, Appendix A). ANCOM-BC showed no significant differences between EM-risk and EM non-risk groups (Appendix A).

### 3.4. Comparisons of Gut Microbiota Structure between EF-Risk and EF-Control Groups

In this section, we present the results of the comparisons of the gut microbiota composition between the EF-risk and control groups. As described in Section 2.2.1, the EF-control group comprised children who were not at risk in any of the three indexes (ICS, FL, and MC). We then checked for any discrepancies by comparing these results with those from the non-risk group, which included children who showed risk in indicators other than the targeted one, in Section 3.1, Section 3.2 and Section 3.3. As demonstrated in Figure 5, Figure 6 and Figure 7 and Appendix A, none of the results of the gut microbiome analysis between the EF-risk and control groups differed from those reported in Section 3.3. Therefore, it was confirmed that even if children who fell under risks other than the target were included in the non-risk group, it did not affect the analysis of the gut microbiota.

### 3.5. Post Hoc Analysis of Differences between ISC-Risk and Non-Risk Groups

Among the three broad indices, the ISC-risk and ISC non-risk groups showed clear differences in microbiome composition. Based on the findings, we conducted a post hoc analysis to examine whether the two groups showed distinctive dietary habits and physical symptoms.

#### 3.5.1. Dietary Habits

The Brunner–Munzel test revealed that the ISC-risks consumed green and yellow vegetables significantly less frequently than the ISC non-risks (*p* = 0.002, *q* = 0.026) (Figure 8a, Appendix A). Moreover, the Fisher’s exact test revealed that the proportion of picky eaters in the ISC-risk group was significantly higher than that in the ISC non-risk group (*p* = 0.002, *q* = 0.026) (Figure 8b, Appendix A). Finally, we found no significant differences in intake frequencies of the other 23 food items or JDI scores between the groups (Appendix A).

#### 3.5.2. Physical Symptoms

The comparison of diarrhea frequency using the Mann–Whitney *U* test showed no difference between the ISC-risk and non-risk groups. Fisher’s exact test revealed no differences in abnormal stool form or allergy symptoms (urticaria/atopy, asthma, allergic rhinitis, hay fever, food allergy; see Appendix A for details) between the groups.

For detailed results on dietary habits and physical symptoms in relation to FL and EM, see Appendix A.

## 4. Discussion

We focused on the gut microbiota as a potential biological factor that might influence EF development in early childhood. Specifically, we hypothesized that of the two types of EF, EER might be more closely associated with some characteristics of the child gut microbiota. We found that children who exhibited developmental risk for ISC (which reflects EER), demonstrated a significantly higher relative abundance of the genera *Actinomyces* and *Sutterella* compared to those of the ISC non-risk group. Importantly, in contrast to ISC, the children at risk for problems with FL (involved in EER) and EM (involved in CC) had no distinctly different gut microbiome from the respective non-risk groups. These results supported our hypothesis that one component of EER, ISC, is significantly associated with the gut microbiota.

As described above, the ISC risk group had higher relative abundances of *Actinomyces* and *Sutterella* compared to the ISC non-risk group. Previous studies have suggested that relative abundance of these organisms is positively correlated with inflammatory cytokines (i.e., protein levels of interleukin-8, tumor necrosis factor-α, C-reactive protein) [64,65,66]. Furthermore, clinical studies have demonstrated that patients diagnosed with inflammatory bowel disease have higher relative abundances of *Actinomyces* and *Sutterella* compared to non-risk groups, suggesting a potential association between these genera and gastrointestinal tract inflammation [67,68]. Therefore, children whose ISC development is at risk may have a higher abundance of inflammation-associated bacteria (i.e., dysbiosis). 

In contrast, alpha and beta diversity did not differ between the two ISC development groups. Previous research has reported inconsistent results concerning whether alpha diversity is negatively associated with negative emotionality or fear reactivity at 2.5 months of age [25]. However, it is important to consider the dynamic development of the gut microbiota during the early stage of life. The composition and diversity of the gut microbiota changes drastically during the first few years; diversity approaches adult levels by around three years of age [16,17]. The relationship between the development of the gut microbiome and mental function may fluctuate and change dynamically from infancy to early childhood.

Additionally, the children at risk for EM and FL development showed no significant differences in the gut microbiota with their non-risks. Several studies have suggested that gut microbiota composition during infancy is associated with functional connectivity and structure of the limbic system [21,22]. In a study on adults, there was evidence of a closer relationship between the gut microbiota and limbic system activities involved in processing emotion through probiotic interventions [23]. Our results are consistent with these findings. As the EM index indicates the CC ability to solve non-emotional problems, its association with the gut microbiota in the current study was not strong. Another reason for the results of the present study, where the association with EER was more evident than that with CC, could be attributed to the fact that the prefrontal cortex is in an immature stage in early childhood, and thus has difficulty regulating the limbic system [18,19,20]. However, further research is required to explain why the FL risk, relevant to emotional switching including in the EER domain, showed no association with the gut microbiota.

To investigate the potential ecological factors affecting child gut microbiome composition, we conducted post hoc analyses focusing on their dietary habits. ISC-risk children consumed fewer green and yellow vegetables and included approximately twice as many picky eaters (in terms of percentages) compared to ISC non-risk children. This result is consistent with research showing that young children who are picky eaters particularly avoid vegetables, leading to lower fiber intake [69]. Importantly, higher fiber intake is associated with a lower abundance of bacteria associated with inflammation (e.g., *Actinomyces*) and a higher abundance of short-chain fatty acid (SCFA)-producing bacteria (e.g., *Lachnospira*), which promote intestinal tract homeostasis [70,71]. Therefore, the dietary habits of children at ISC-risk might be related to increased levels of inflammation-related bacteria, and dietary interventions aimed at balancing the gut microbiota may be an effective method for supporting ISC development.

We found no differences in physical symptoms between the ISC-risk and non-risk groups. Gut microbial changes (i.e., dysbiosis) may be a factor in childhood allergic diseases [72]. Actually, children with allergies such as food allergy and asthma have a higher abundance of inflammation-associated bacteria and a lower abundance of SCFA-producing bacteria [73,74]. In this study, although ISC-risk children were observed to have dysbiosis (characterized by increased inflammation-related bacteria), they did not show more physical symptoms such as allergies. However, the relationship between ISC risk and physical symptoms may change over time with temporal changes in gut microbiome composition; therefore, the relationship should be monitored longitudinally at multiple developmental points, including during school age and adulthood.

This study has several limitations. First, because of the cross-sectional design, we do not yet know whether our findings are relevant to long-term developmental outcomes, or whether improving the gut microbiota through dietary habits or other interventions (e.g., prebiotics) reduces EF risks. Longitudinal cohort studies and intervention studies are required to clarify how dysbiosis or changes in the gut microbiota during early childhood influence the risk of EF. Second, we used the BRIEF-P questionnaire to assess children’s EF risks. This measurement tool is based on subjective responses by caregivers, whose mood and beliefs about their children might influence their answers. Validation by observing and assessing child behaviors directly would be an important methodological improvement. Despite the above limitations, our study serves as a preliminary step toward understanding the relationships among three factors: gut microbiota, dietary habits, and executive functions. To enhance our understanding of these relationships, especially when considering dietary interventions targeting gut microbiota, it is evident that the method of assessing dietary habits needs to be refined, extending beyond the frequency of intake of each food category to include a detailed examination of nutritional intake in future research.

## 5. Conclusions

The current study revealed that the gut microbiome is associated with the development of EF from 3 to 4 years of age, especially in the specific domain of EF, EER. This time period is crucial in development because EF develops dramatically, and the gut microbiome composition attains an adult-like configuration during this period. It is possible that the child microbiome is associated with variations in EF development. Many EF intervention programs have been proposed to reduce problematic behaviors and improve cognitive abilities in early childhood. However, our results show that more “individualized” intervention methods are necessary to address risks to children’s cognitive development, and to protect life-long physical and mental health. We believe that focusing on child gut microbiota and dietary habits would be valuable for designing new and effective intervention strategies for diverse children.

## Figures and Tables

**Figure 1 microorganisms-11-02245-f001:**
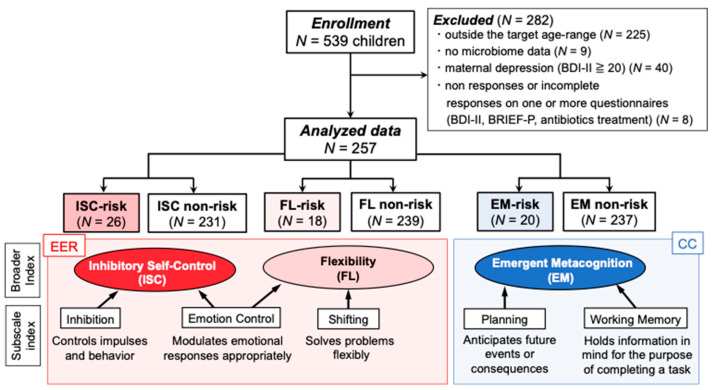
Participant enrollment based on BRIEF-P scores. Note. Each non-risk group included children with an EF risk different from the target risk. The detailed breakdown of type of risk and the number of children in each inhibitory self-control (ISC), flexibility (FL), and emergent metacognition (EM) risk/non-risk group is shown in Appendix A.

**Figure 2 microorganisms-11-02245-f002:**
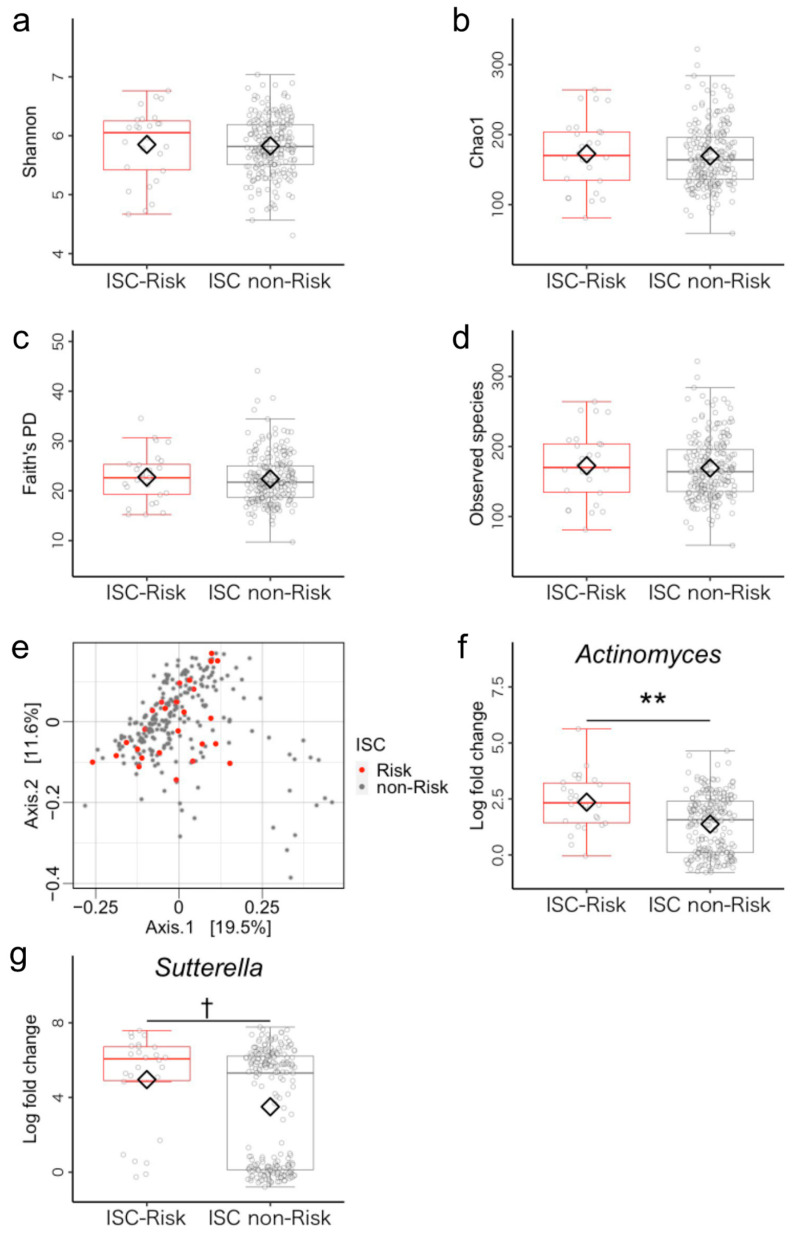
Gut microbiota community structure in the inhibitory self-control (ISC)-risk and non-risk groups. (**a**–**d**) Comparison of alpha diversity. Statistical significance was determined using covariance and (**e**) beta diversity analysis. The dissimilarity of bacterial communities was analyzed using principal coordinate analysis (PCoA) based on Bray–Curtis metrics. (**f**,**g**) Comparison of relative abundance of genus-level bacteria (log-fold change). The compositions of the microbiomes were analyzed using bias correction (ANCOM-BC). Note. Antibiotic treatment within the past three months and sex were used as covariates. ** *q* < 0.01; ^†^ *q* < 0.10.

**Figure 3 microorganisms-11-02245-f003:**
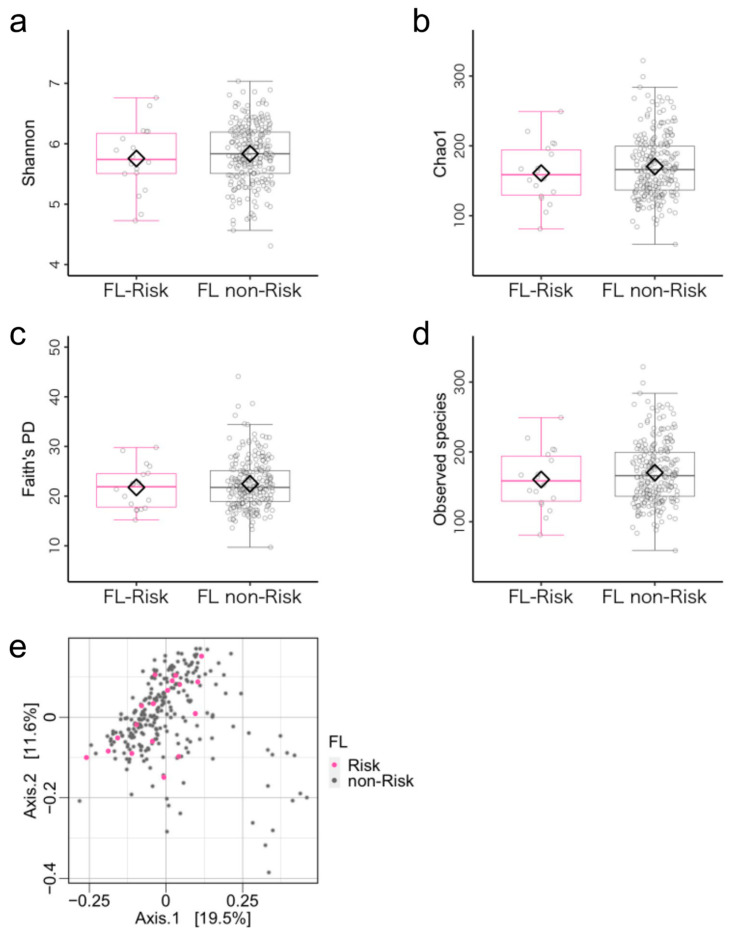
Gut microbiota community structure in the flexibility (FL)-risk and non-risk groups. (**a**–**d**) Comparison of alpha diversity. Statistical significance was determined using covariance and (**e**) beta diversity analysis. The dissimilarity of bacterial communities was analyzed using principal coordinate analysis (PCoA) based on Bray–Curtis metrics. Note. Antibiotic treatment within the past three months and sex were used as covariates.

**Figure 4 microorganisms-11-02245-f004:**
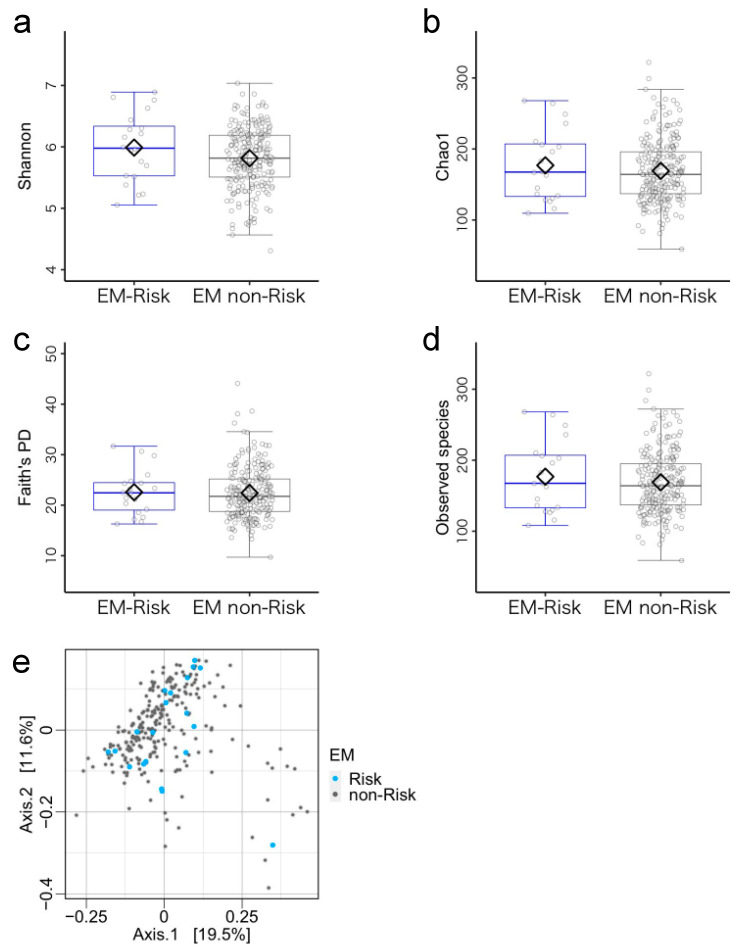
Gut microbiota community structure in the emergent metacognition (EM)-risk and non-risk groups. (**a**–**d**) Comparison of alpha diversity. Statistical significance was determined using covariance and (**e**) beta diversity analysis. The dissimilarity of bacterial communities was analyzed using principal coordinate analysis (PCoA) based on Bray–Curtis metrics. Note. Antibiotic treatment within the past three months and sex were used as covariates.

**Figure 5 microorganisms-11-02245-f005:**
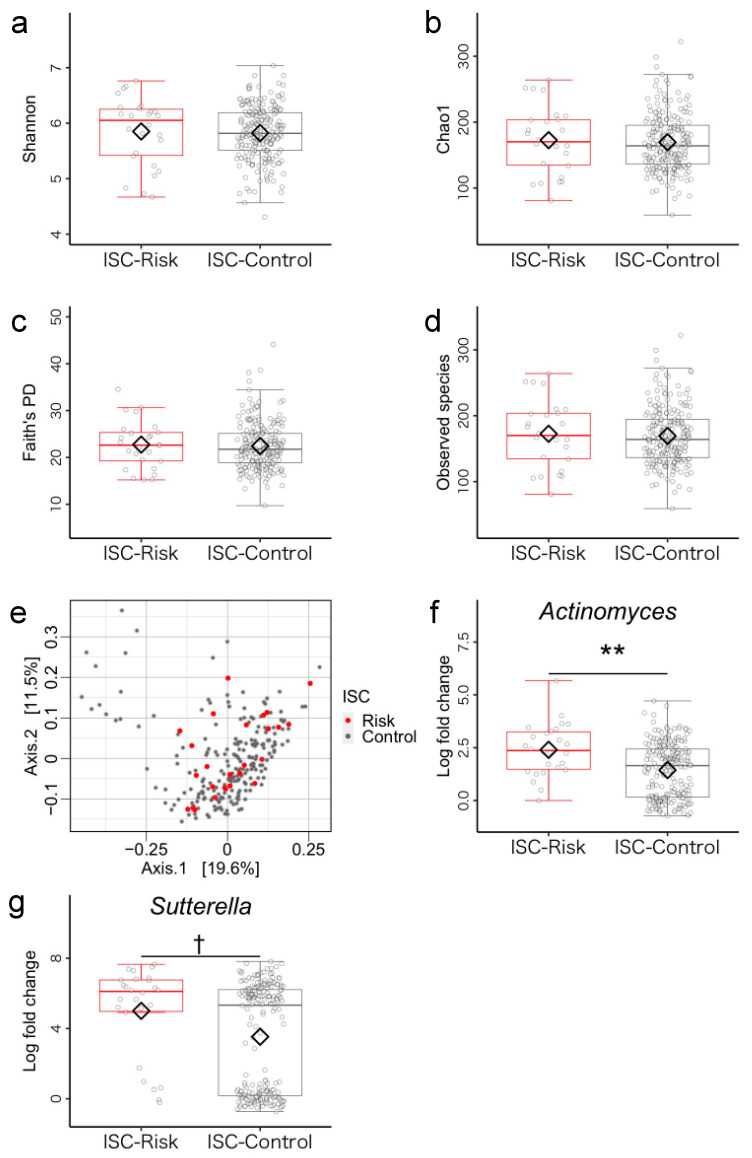
Gut microbiota community structure in the inhibitory self-control (ISC)-risk and control groups. (**a**–**d**) Comparison of alpha diversity. Statistical significance was determined using covariance and (**e**) beta diversity analysis. The dissimilarity of bacterial communities was analyzed using principal coordinate analysis (PCoA) based on Bray–Curtis metrics. (**f**,**g**) Comparison of relative abundance of genus-level bacteria (log-fold change). The compositions of the microbiomes were analyzed using bias correction (ANCOM-BC). Note. Antibiotic treatment within the past three months and sex were used as covariates. ** *q* < 0.01; ^†^ *q* < 0.10.

**Figure 6 microorganisms-11-02245-f006:**
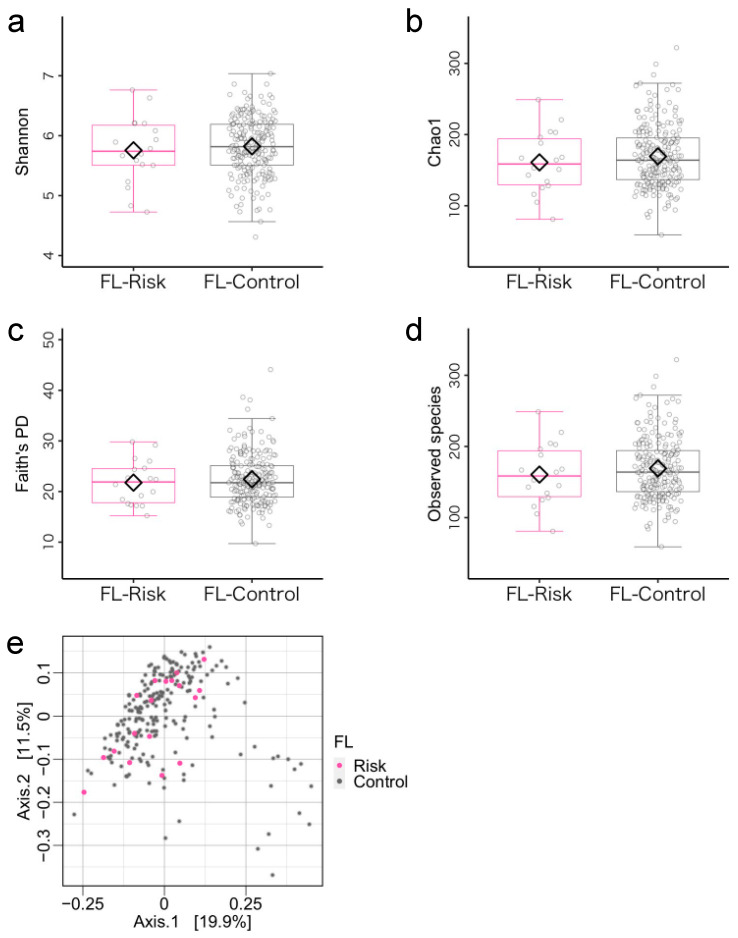
Gut microbiota community structure in the flexibility (FL)-risk and control groups. (**a**–**d**) Comparison of alpha diversity. Statistical significance was determined using covariance and (**e**) beta diversity analysis. The dissimilarity of bacterial communities was analyzed using principal coordinate analysis (PCoA) based on Bray–Curtis metrics. Note. Antibiotic treatment within the past three months and sex were used as covariates.

**Figure 7 microorganisms-11-02245-f007:**
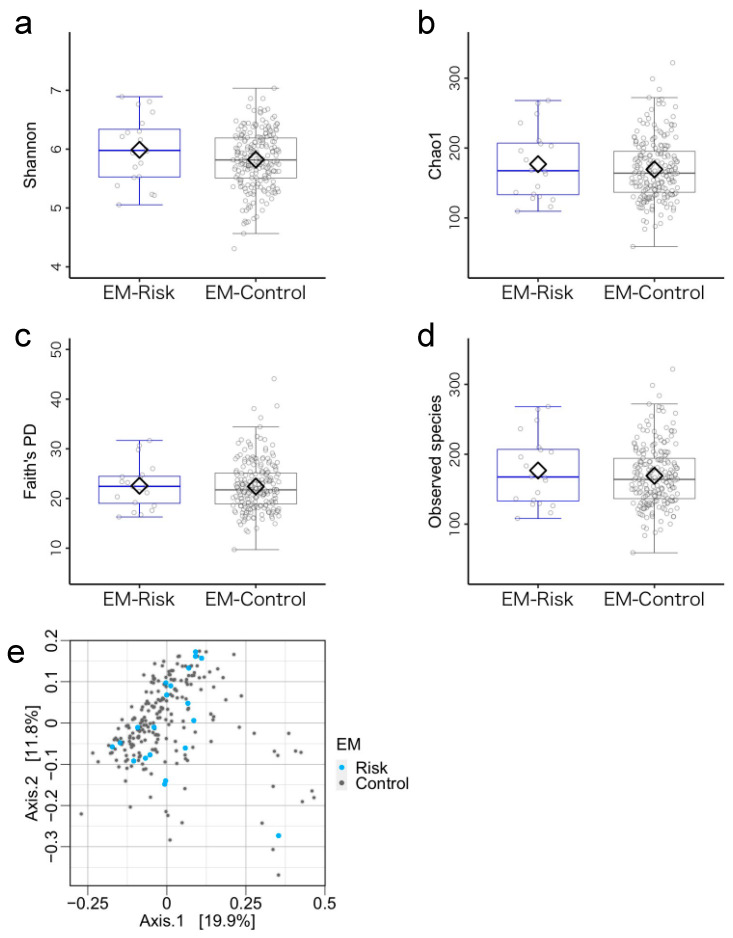
Gut microbiota community structure in the emergent metacognition (EM)-risk and control groups. (**a**–**d**) Comparison of alpha diversity. Statistical significance was determined using covariance and (**e**) beta diversity analysis. The dissimilarity of bacterial communities was analyzed using principal coordinate analysis (PCoA) based on Bray–Curtis metrics. Note. Antibiotic treatment within the past three months and sex were used as covariates.

**Figure 8 microorganisms-11-02245-f008:**
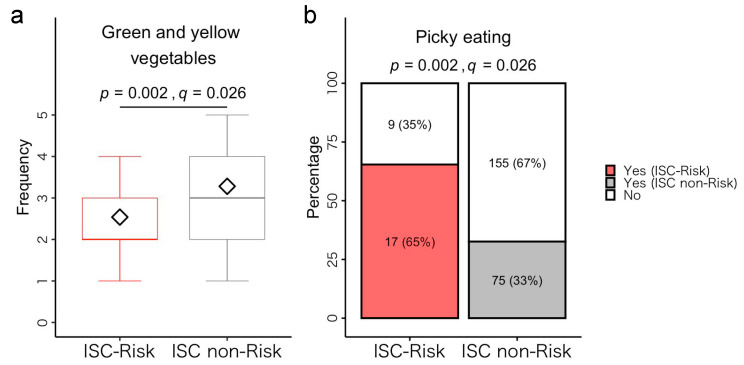
Comparison of intake frequency of green and yellow vegetables (**a**) and the proportion of picky eaters (**b**) between the ISC-risk and non-risk groups.

## Data Availability

The studies presented here were not preregistered. Readers may access the code for the analytical routines on OSF (https://osf.io/qh6fw/files/osfstorage). The data can be requested via contact the corresponding authors after publication.

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
