# Peer review of "Altered Gut Microbiota Composition Is Associated with Difficulty in Explicit Emotion Regulation in Young Children"

_microorganisms, 2023, doi:10.3390/microorganisms11092245_

Round 1
Reviewer 1 Report
Executive function (EF), encompassing explicit emotion regulation (EER) and cognitive control (CC), is vital for lifelong physical and mental health. However, factors influencing EER development remain unclear. The gut microbiota has emerged as a potential biomarker for adult health risks, yet its role in early childhood is poorly understood. The authors investigated 257 children aged 3-4 to explore the association between gut microbiota and EER risk, and revealed a higher abundance of Actinomyces and Sutterella genera in children at EER risk, suggesting the involvement of the gut microbiome in childhood EF development, specifically in EER. This manuscript was well written and structured, their findings are straightforward and significant for uncovering the role of gut microbiota in the development of EF. I only have some minor comments.
Line 150. The full spelling of Inhibitory Self-Control (ISC), Flexibility (FL), and Emergent Metacognition (EM) should be presented in the figure’s legend.
Line 213. The sequence processing is crucial for repeating the analysis, so it would be better to show the details here.
Line 274. The data in Table S16 would present in a graph in the main manuscript.
I would recommend a PCoA plot to show the beta diversity in this manuscript.
Reviewer 2 Report
The paper is interesting but some minor concerns should be considered
1-it seems that the abstract is not providing sufficient information about the work that has been done. To improve it, I will expand the length of the abstract and ensure that it includes the main points such as the background of the study, the methods used, the results obtained, and the final conclusions drawn from the research.
2- I detected a few small errors with the English in the text. please edit the manuscript once more.
need to revise again
Reviewer 3 Report
In this article, Fujihara et al. reported how altered gut microbiota composition is associated with difficulty in explicit emotion regulation in young children. The authors have collected so much information and well analyzed the data. However, the article does not give a clear view of the association between the change in gut composition and EER risk. In several instances, it lacks precision and specificity in the description. For example,
Line 87-88: why authors hypothesized that “CC-related links would be less clear” before conducting the study?
The selection and no risk assessment criteria for controls for every study in the article are not clear. Too many parameters have been included in the study which is making it harder to understand the selection of children for a particular study and its respective controls. The complex selection parameters could hinder conclusive data.
The results in the main text are mentioned in two figures and are not explained well. All data is shown in the supplementary material. Some data should be shown in the main text to support the interpretation of the results. The article is about “altered gut microbiota composition” so, the results of this section must be added to the main text instead of adding all data in the supplementary material. The main review point is to reshape the article by making some clear figures to explain the results of different sections. How dietary habits and physical symptoms affected the gut microbiota in children and what authors suggest to control the EER risk in children?
What is the probability of change in gut microbiota and its associated EER risk when these children will grow a bit more and have different eating habits than at the time of study?
In Table S6, mention what is “beta”, “SE” and “W” in the legend.
Please consider revising some points which have been mentioned in the attached PDF of the manuscript.
The article needs English editing to remove general grammar mistakes

The article needs English editing to remove general grammar mistakes
Round 2
Reviewer 3 Report
The authors have addressed the comments. This work sheds light on the significance of the development of eating habits and healthy gut microbiota at the early stage of life to avoid mental disorders.
Minor English editing should be done.